# Community-based conservation with formal protection provides large collateral benefits to Amazonian migratory waterbirds

João Vitor Campos-Silva[1,2]*, Carlos A. Peres[3], Joseph E. Hawes[1,4], Mark I. Abrahams[5], Paulo C. M. Andrade[6], Lisa Davenport[7,8]

1 Faculty of Environmental Sciences and Natural Resource Management, Norwegian University of Life Sciences, Ås, Norway, 2 Institute of Biological and Health Sciences, Federal University of Alagoas, Maceió, AL, Brazil, 3 School of Environmental Sciences, University of East Anglia, Norwich Research Park, Norwich, United Kingdom, 4 Applied Ecology Research Group, School of Life Sciences, Anglia Ruskin University, Cambridge, United Kingdom, 5 Field Conservation and Science Department, Bristol Zoological Society, Bristol, United Kingdom, 6 Departamento de Produção Animal e Vegetal, Laboratório de Animais Silvestres, Universidade Federal do Amazonas, Manaus, AM, Brazil, 7 Department of Biology and Florida Museum of Natural History, University of Florida, Gainesville, FL, United States of America, 8 College of Science and Engineering, James Cook University, Cairns, Queensland, Australia

* jvpiedade@gmail.com

**Data Availability Statement:** All relevant data are within the manuscript and its Supporting Information files.

## Abstract

Populations of migratory waterbirds are facing dramatic declines worldwide due to illegal hunting, habitat loss and climate change. Conservation strategies to reverse these trends are imperative, especially in tropical developing countries, which almost invariably allocate insufficient levels of investment for environmental protection. Here, we compared the effectiveness of sustainable-use Protected Areas (PAs) and Community-based Conservation (CBC) arrangements for the conservation of migratory waterbirds that breed on seasonal riverine sandy beaches in Brazilian Amazonia. We modeled local population responses of four migratory waterbird species on 155 beaches along a ~1,600 km section of a major tributary of the Amazon, as a function of community enforcement, official protection status, human pressure and landscape features. We show that 21 community-protected beaches within the study area host more than 80% of all sampled birds. Black Skimmers showed the most dramatic response, with breeding numbers 135-fold larger in CBC arrangements compared to beaches with no official protection status. The same pattern was observed for nesting Large-Billed and Yellow-Billed Terns. For the Near Threatened Orinoco Goose, PA status was the strongest predictor of local population size. These dramatic results demonstrate the value of protected refugia, achieved through the concerted action of participating local communities, to support breeding populations of key waterbird species. This highly-effective and low-cost conservation model can potentially be replicated in other regions of the developing world experiencing increasingly intensive exploitation of riverine natural resources.

**Funding:** This study was funded by a Darwin Initiative for the Survival of Species grant (DEFRA, UK; Ref. 20-001) awarded to C.A.P., a CAPES Ph.D. scholarship (Ref. 1144985) and a postdoctoral grant to JVC-S (no. 295650) through the Belmont Forum and BiodivERsA joint call for research proposals, under the BiodivScen ERA-Net COFUND programme, and with the funding organisations French National Research Agency (ANR), São Paulo Research Foundation (FAPESP), National Science Foundation (NSF), the Research Council of Norway and the German Federal Ministry of Education and Research (BMBF), a CAPES postdoctoral grant (Ref. 1530532) and internal funding from Anglia Ruskin University to J.E.H., and a National Geographic Society grant (Ref. WW1-277R-18) to L.D. The funders had no role in study design, data collection and analysis, decision to publish, or preparation of the manuscript.

**Competing interests:** The authors have declared that no competing interests exist.

## Introduction

Freshwater wetlands in the tropics represent one of the world's most threatened ecosystems, with higher rates of species loss compared with terrestrial environments, rendering their conservation an imperative [1–3]. In Brazilian Amazonia, for example, aquatic environments are severely threatened by numerous anthropogenic activities, including deforestation, overexploitation, pollution, mining and large hydropower infrastructure [4]. Although Brazil has developed one of the most comprehensive protected area (PAs) systems on Earth, with 2201 PAs covering more than 250 million hectares [5], it is nevertheless inadequate to curb most of these impacts [4]. In this context, new conservation strategies that align the protection of biodiversity and local livelihoods must be implemented.

Freshwater turtles (*Podocnemis* spp.) are one of the most emblematic taxa in Amazonian freshwater systems, and have been a pivotal resource for traditional societies from pre-Columbian to modern times [6]. Turtle meat and eggs provide protein, fuel oil and medicine, and the carapace is used in rituals and for manufacturing tools [7–9]. During the last century, however, wild populations were decimated in many Amazonian rivers due to overexploitation [10]. Faced with dramatic population declines, especially of the South American River Turtle (*P. expansa*), Community-Based Conservation (CBC) arose to reverse the depletion of this high-value resource, ensuring the protection of breeding sites, which are usually on large sandy beaches where turtle harvesting is not allowed [11]. Currently in Brazil, there are approximately 390 seasonal nesting beaches protected by local communities, which utilize an effective engagement and surveillance regime, in an intensive population recovery effort for freshwater turtle [12].

Community protection of sandy beaches in Amazonian rivers occurs during the dry season, which lasts for approximately five months. During this period, local beach guards are stationed in small wooden huts built on riverbanks in front of the beach. A rotation of beach guards maintains constant vigilance during this period. The pressure from poachers is very high due to the high commercial value of freshwater turtles. Beach guards can receive a small payment from a multi-partner arrangement, usually delivered in the form of food items [12], but in most places all work is voluntary. The permanent physical presence of beach guards greatly increases local governance and effectiveness of beach protection, delivering indirect conservation benefits for a wide range of vertebrate and invertebrate taxa [12]. Collateral benefits from CBC protection of umbrella species such as *Podocnemis* spp. have previously received little attention in the literature, but birds that depend on seasonal sandy beaches for feeding or reproduction may benefit from the protection afforded to turtle-nesting beaches, probably because of the low level of threats, including egg-collecting, agriculture and fishing in protected beaches [12,13]. Most of these species are migratory waterbirds that are also facing dramatic declines worldwide due to illegal hunting, habitat loss and climate change [14], especially in developing countries with weak enforcement of conservation regulations [15].

The conservation of migratory birds has largely depended on Protected Areas (PAs) to mitigate the impact of habitat loss and increase habitat connectivity to overcome potential migratory barriers [16,17]. PAs usually play a critical role in protecting breeding sites [18], although in the case of "paper parks" where true enforcement is lacking, effective protection typically fails [19]. In addition, only around 9% of migratory bird species worldwide are adequately protected by PAs, when considering protection needs across their full annual cycles [20]. In this context, it is crucial to implement long-term monitoring of waterbird populations to understand the spatial dynamic of each species and their population trends under different protection scenarios [21,22]. This information is crucial to inform large-scale environmental

policies, increasing and strengthening conservation strategies both inside and outside PA boundaries.

The occurrence and abundance of migratory waterbirds on floodplains environments are influenced by both biotic and abiotic characteristics [23,24]. For example, the choice of individual beaches for breeding can often be predicted by environmental and physical factors, including beach size, geographic isolation, substrate type, and distance to large colonies of the same species [25]. However, the protection status of sandy beaches appears to be an important factor that can frequently override environmental and physical factors, since well-protected beaches host larger populations with higher reproductive success [12].

In this study, we targeted four species of migratory waterbirds: the Black Skimmer (*Rynchops niger*); Large-Billed Tern (*Phaetusa simplex*); Yellow-Billed Tern (*Sternula superciliaris*); and Orinoco Goose (*Neochen jubata*). Black Skimmers complete impressive trans-Andean migrations, with some individuals traveling down the Pacific Coast and staying through the austral summer in the Gulf of Arauco, Chile [26]. The species usually nests with Large-Billed and Yellow-Billed Terns on sandy beaches and sandbars of rivers and lakes [27,28]. The formation of multi-species colonies can increase the reproductive success of colonial breeders through collective anti-predator defense, even if species do not accrue equal benefits [27,29]. The Near-Threatened Orinoco Goose also occurs on river margins, using sandbank habitats for foraging and resting. Due to its large body size and crop-damaging potential, the Orinoco Goose is often hunted by local residents [30].

We conducted a comprehensive assessment of 155 beaches spread along a ~1,600 km fluvial section of the Juruá River in western Brazilian Amazonia, examining the effect size of PAs and CBC arrangements on the occurrence and abundance of waterbird populations, controlling for other important factors such as anthropogenic pressure and landscape features. We hypothesize that fluvial beaches guarded by local communities to protect freshwater turtles also provide strong benefits for the conservation of Amazonian migratory waterbirds. Our results increase the understanding of the role of sustainable-use protected areas and community-based conservation regimes on migratory bird conservation. This is important because empowering local communities to protect their territory has become an effective tool to ensure the protection of different taxonomic groups that inhabit the threatened Amazonian floodplains [12,31], and could be an additional strategy to protect migratory birds.

## Materials and methods

### Study area

This study was conducted on 155 fluvial beaches (mean ± SD: arc length = 2,238 ± 940.6 m, area = 23.3 ± 15.4 ha) along ~1,600 km of the Juruá River (5˚ 2'32.71"S; 66˚58'19.93"W), a highly productive major tributary of the Amazon River (Fig 1). The Juruá landscape is comprised of seasonally-flooded (várzea) forests within the floodplain, and upland (terra firme) forests that are not inundated [32]. The dry and wet seasons coincide with periods of low- (August–November) and high-water levels (January–June), with a pronounced flood pulse often exceeding 10m in amplitude [33], which strongly impacts biological communities [34]. During the dry season, convex sandy point bars (beaches) are formed along large sections of the main meandering river channel, creating suitable nesting habitat for several taxonomic groups, including migrant waterbirds [12]. The Juruá River also hosts three sustainable-use PAs, aiming to ensure the protection of biodiversity and local livelihoods (Fig 1). The Médio Juruá Extractive Reserve (RESEX Médio Juruá) was created in 1997 and encompasses 253,227-hectares. This reserve is legally occupied by some 700 people distributed across 13 villages. The Lower Juruá Extractive Reserve (RESEX Baixo Juruá) was created in 2001 and

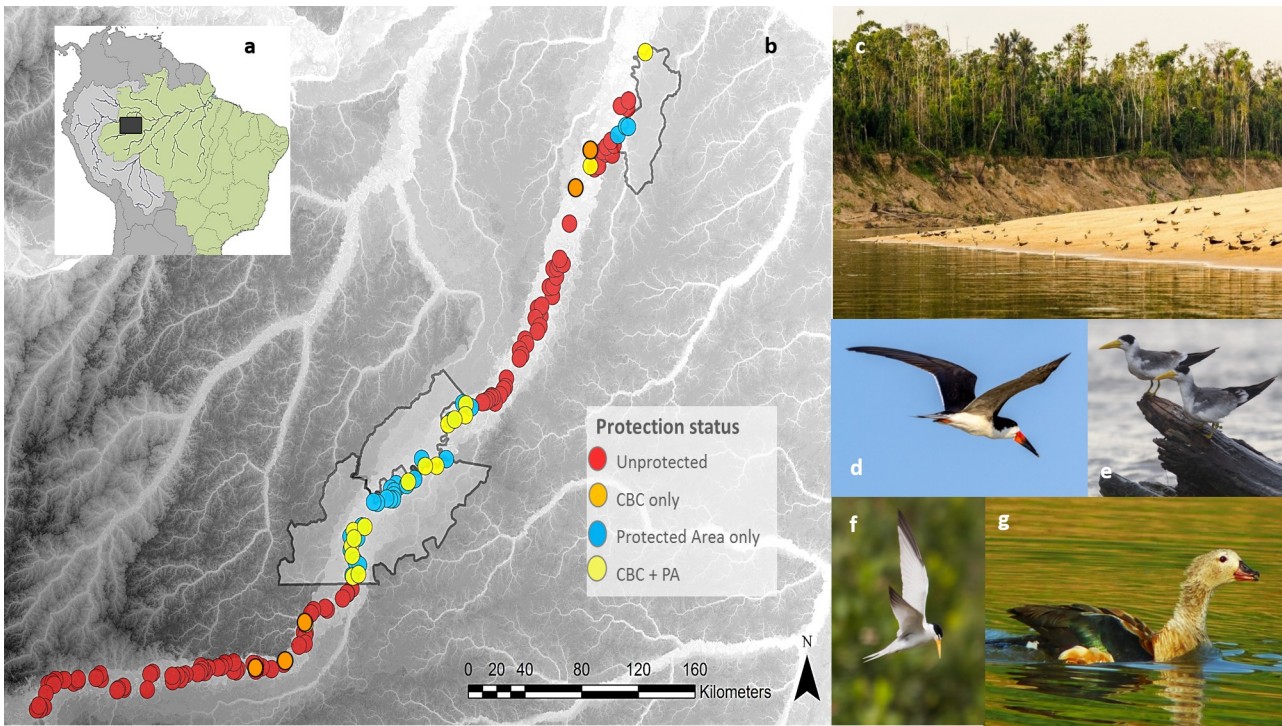

**Fig 1. Map of surveys for four waterbird species along the Juruá River in western Brazilian Amazonia.** (A) Study location in western Brazilian Amazonia; showing (B) 155 sandy beaches located along ~1600 km of the Juruá River. Red, orange, blue and yellow circles indicate the protection status of beaches corresponding to: Unprotected; outside formal PA but under CBC (CBC-only); inside formal PAs but without CBC (PA-only); and, inside formal PAs and under CBC (CBC + PA), respectively; solid black lines represent the boundaries of three sustainable-use PAs. (C) example of a sandy beach guarded under a CBC arrangement; (D) Black Skimmer (*Rynchops niger*); (E) Large-Billed Tern (*Phaetusa simplex*); (F) Yellow-Billed Tern (*Sternula superciliaris*) and (G) Orinoco Goose (*Neochen jubata*). Landsat-7 image courtesy of the U.S. Geological Survey.

encompasses 187,981 ha and 1,380 people. Finally, the state-managed Uacari Sustainable Development Reserve (RDS Uacari) was created in 2005 where ~1,200 people live in 32 villages across 632,949 ha. The local economy is sustained by fisheries, slash-and-burn cassava agriculture, and non-timber forest products such as oil seeds and palm fruits [35].

Along the Juruá River, community-protection of some beaches was initiated by local rubber tappers in the early 20th century to provide meat and eggs to owners of natural rubber stands. After the decline of rubber activity, protection of turtle spawning beaches was stimulated by the Brazilian environmental agency (IBDF/IBAMA and Projeto Quelônios da Amazônia), when it became the responsibility of local communities through the implementation of CBC initiatives [11,13]. This program is supported by a multi-partner arrangement, where government, local associations, NGOs, universities and local communities share the challenge of keeping the program running. Considering the entire Juruá basin, there are currently 25 beaches protected by ~ 62 beach-guards (2–4 per beach). The CBC arrangement occurs inside or outside formal PAs, and currently there are 19 community-protected beaches inside PAs and six outside. Although some fluvial beaches in Amazonia have been protected for longer, most of the CBC arrangements in our study area were only established in the last decade; length of protection ranges between 2 and 43 years. The beaches selected for CBC protection reflect the feasibility of regular visitation by local monitors, and these beaches are therefore not necessarily the most ecologically suitable sites, or those used historically by turtles or migratory birds. Indeed, these beaches typically had depleted turtle and bird populations prior to the initiation of CBC protection.

## Waterbird surveys

Censuses of beach-dependent waterbirds along the Juruá River were conducted slowly on foot (mean velocity ~1.25 km.h–1) by an experienced observer, censusing a pre-determined linear transect through the middle of the beach [36]. Census walks started after dawn at 06:30h and did not finished later than 10:00h. We counted the total number of individuals of each species using $10 \times 40$ binoculars, sampling up to 6 beaches per day, and the counts were usually completed between 20 to 90 min, depending on beach area. Bird counts were conducted during the dry season from the 2nd to the 31st of August 2016, with one survey per beach. On beaches occupied by a large single- or multi-species group of birds (e.g. more than 20 individuals) the observer repeated the count three times, using the smallest number of individuals as a conservative estimate. We also maintained a sufficient distance to prevent flushing waterbirds during our surveys, and to avoid the risk of double counting. This study was authorized by the Brazilian Government through the Instituto Chico Mendes de Conservação da Biodiversidade (ICMBio; permit number 71753).

## Data analysis

Our study design comprises 155 beaches under four different levels of protection status: (i) CBC-only = outside a PA and protected by a local community; (ii) PA-only = inside a formal PA but without community protection; (iii) PA + CBC = inside a PA and protected by a local community; or (iv) unprotected = outside a PA and without community protection. Most beaches protected by local communities are clustered within the middle portion of the Juruá River, due to the strong social organization within this particular area that incentivizes local communities to focus on the conservation of freshwater turtles [12]. We therefore used Moran's *I* to test for potential spatial autocorrelation in our dataset. We used a matrix of inverse distance weights in the *ape* package [37], to calculate Moran's *I*, which varies between 1 and -1, where positive autocorrelation represents positive values of *I*, negative autocorrelation represents negative values, and values close to zero represent no autocorrelation [38]. We found no autocorrelation for our target species (*Rynchops niger*: *I* = -0.01; *Phaetusa simplex*: *I* = -0.04; *Sternula superciliaris*: *I* = 0.1; *Neochen jubata*: *I* = -0.1), and therefore did not include any steps to account for spatial autocorrelation in our subsequent models.

We performed general linear models (GLMs) to examine the relative strength of protection status on the abundance of migratory waterbirds, as the number of individuals per hectare of each species. In addition to protection status as the explanatory variable, we controlled for the following patch and landscape covariates: *beach area*, calculated as the total area of each beach in hectares, using the extreme geo-referenced points obtained by Global Positioning System (GPS) along the convex river meander and measuring its maximum width; *distance to nearest community* and *distance to nearest town*, both measured with GPS units as the nonlinear fluvial distance by boat and extracted for each beach using ArcGIS (version 10.2). All datasets are available in the Supporting Information (S1 Table). We recognize that time of day could potentially influence the behavior of our target species [39]. As we did not find any significant difference between early morning and later morning surveys, we did not include this factor in our subsequent models (S1 Fig).

Models were fitted with lmer in the *lme4* package and every model combination was examined using the *MuMIn* package [40]. We selected the most parsimonious models based on the lowest Akaike Information Criterion corrected for small sample sizes (AICc). ΔAICc was calculated as the difference between each model AICc and the lowest AICc, with a ΔAICc < 2 interpreted as substantial support that the model belongs to the set of 'best' models. Akaike weights give the probability that a model is the 'best' model, given the data and the set of

candidate models [41] Following model selection, we performed model averaging, which considers the beta average of all variables included in the most parsimonious models. Explanatory variables were z-standardized to allow comparisons among effect sizes. All assumptions were examined prior to analyses, including linear relations, correlations between explanatory factors, homoscedasticity and distribution of residuals [42] and all analyses were performed within the R platform [43].

## Results

We detected 6,548 individual birds from a total of 155 beaches, including 5 CBC-only (mean area = 24.9 ha, SD = ± 13.8), 27 PA-only (22.1 ha ± 15.9), 14 CBC + PA (38.2 ha ± 21), and 109 unprotected (21.9 ha ± 13.2). CBC beaches, regardless of whether inside or outside formal PAs, hosted far more beach-dependent birds than non-CBC beaches. CBC-only and CBC + PA hosted 1,355 and 4,043 individuals, respectively, meaning that 13.5% of beaches in the census area hosted 82.4% of all individual birds counted (Fig 2). The Large-Billed Tern was the most abundant species (3,028 individuals occurring on 30.3% of sampled beaches), followed by the Black Skimmer (2,531 individuals occurring on 31.6% of sampled beaches), Orinoco Goose (582 individuals occurring on 38% of sampled beaches) and Yellow-Billed Tern (407 individuals occurring on 56.7% of sampled beaches; Table 1).

We found a strong impact of protection status on abundance for most species, where CBC-only and CBC + PA categories showed far more beach-dependent birds than formal PAs and unprotected sites (Fig 3). Therefore, for migratory waterbirds breeding on riverine beaches, the protection delivered by local communities represents the most effective conservation strategy.

The mean population size of Black Skimmer was 148-fold and 136-fold larger on CBC-only and CBC + PA beaches than at unprotected sites, and only 1.3 higher in PA-only beaches compared with unprotected sites (Table 1; Fig 3A). A large percentage (94.5%) of all Black Skimmers were counted on CBC beaches (CBC-only and CBC + PA), which hosted the largest breeding populations (Fig 4A). Modeling Black Skimmer abundance, we found that CBC-only and CBC + PA were the strongest predictors. We found no effect of distance to the nearest rural community or nearest town (Fig 5A).

Large-Billed Tern abundance was 94-fold and 119-fold higher on CBC-only and CBC + PA beaches than unprotected beaches, and 2.4-fold higher in PA-only compared with unprotected beaches (Table 1; Fig 3B); 92.3% of all Large-Billed Terns were on CBC beaches (Fig 4B). Modeling Large-Billed Tern abundance, we found that CBC-only and CBC + PA were also the stronger predictors, followed by distance to the nearest community, which induced a negative response (Fig 5B).

The effect of community protection on Yellow-Billed Tern was also strong, but less pronounced. CBC-only and CBC + PA hosted populations 5.6-fold and 4.8-fold higher compared with unprotected sites while PA-only showed virtually no difference with unprotected sites (Table 1; Figs 3C and 4C). Modelling the abundance of Yellow-Billed Tern, we found that CBC-only and CBC + PA were the only important levels of protection status (Fig 5C).

We highlight that, except for the Orinoco Goose, formal PAs lacking local community engagement do not differ from unprotected sites, in terms of the population sizes of beach-dependent birds. The Orinoco Goose showed a different pattern of beach use, with 60% of all individuals in the PA-only category (Table 1; Figs 3D and 4D). Comparing different categories of protection with unprotected sites, PA-only was the only important protection for abundance of this species (Fig 5D). For this species, we found an abundance 7.6-fold higher in PA-

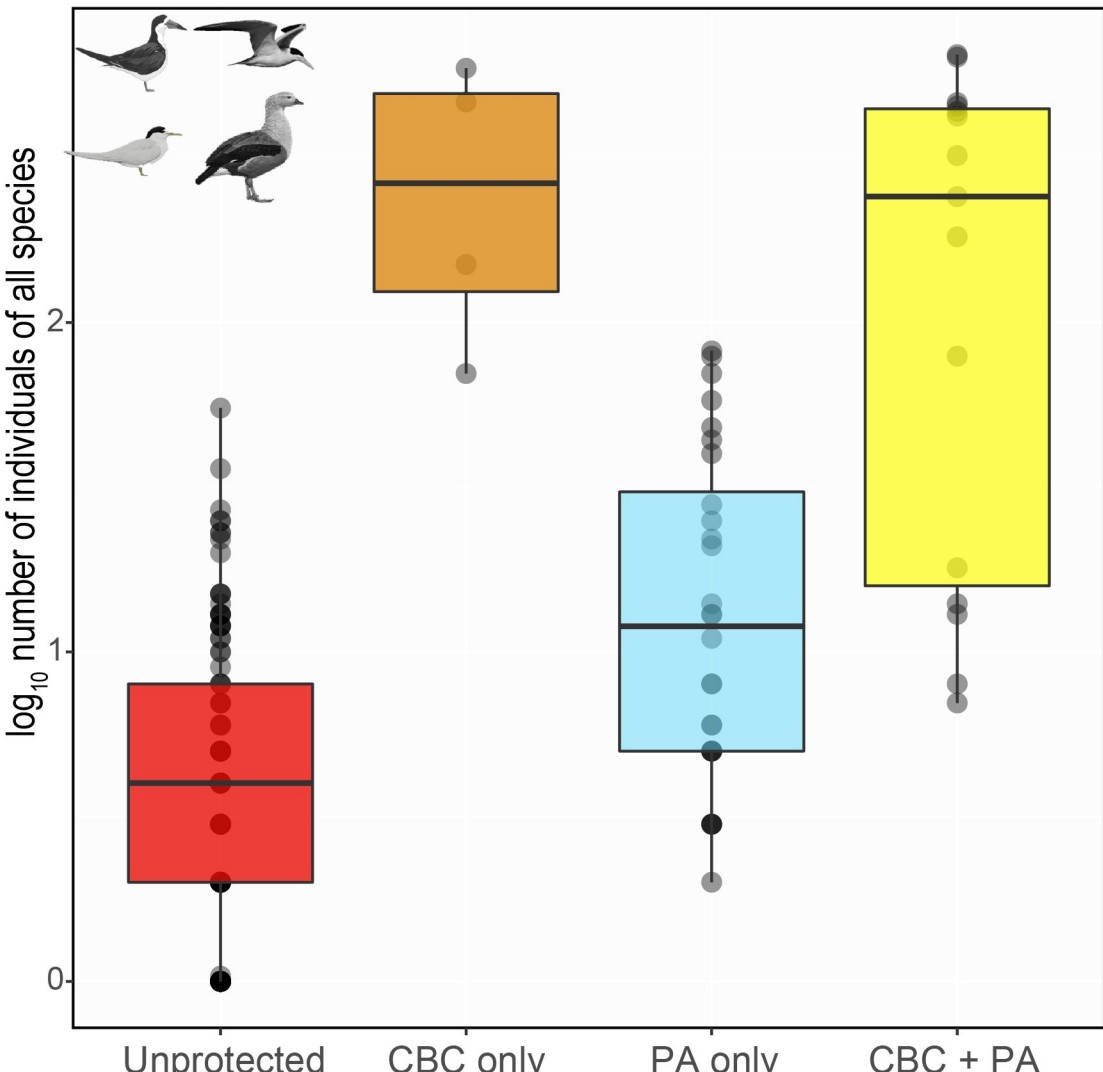

**Fig 2. Box plot showing the total number of waterbird individuals under different protection regimes.** Red, orange, blue and yellow boxes represent beaches that are unprotected, under Community-Based Conservation but outside Protected Areas (CBC-only), inside a PA but without CBC (PA-only), and under CBC inside a PA (CBC + PA), respectively.

**Table 1. Total and mean (± standard deviation) number of individuals for four waterbird species surveyed along the Juruá River, Amazonas, Brazil.**

| | Black Skimmer (*Rynchops niger*) | | Large-Billed Tern (*Phaetusa simplex*) | | Yellow-Billed Tern (*Sternula superciliaris*) | | Orinoco Goose (*Neochen jubuta*) | |
|---|---|---|---|---|---|---|---|---|
| **Protection status** | **Total** | **Mean ± SD** | **Total** | **Mean ± SD** | **Total** | **Mean ± SD** | **Total** | **Mean ± SD** |
| **Unprotected** | 107 | 0.9 ± 54.07 | 147 | 1.3 ± 66.3 | 190 | 1.7 ± 4.4 | 185 | 1.7 ± 10.1 |
| **CBC-only** | 670 | 134.0 ± 64.8 | 615 | 123 ± 79.7 | 48 | 9.6 ± 5.2 | 22 | 4.4 ± 11.7 |
| **PA-only** | 33 | 1.2 ± 69.9 | 85 | 3.1 ± 85.7 | 53 | 1.9 ± 5.6 | 350 | 12.9 ± 12.5 |
| **CBC + PA** | 1721 | 122.9 ± 71.7 | 2181 | 155.8 ± 87.8 | 116 | 8.3 ± 7.3 | 25 | 1.7± 12.5 |

CBC-only, Community-Based Conservation outside Protected Areas; PA-only, inside a Protected Area but without Community-Based Conservation; CBC + PA, Community-Based Conservation inside a Protected Area.

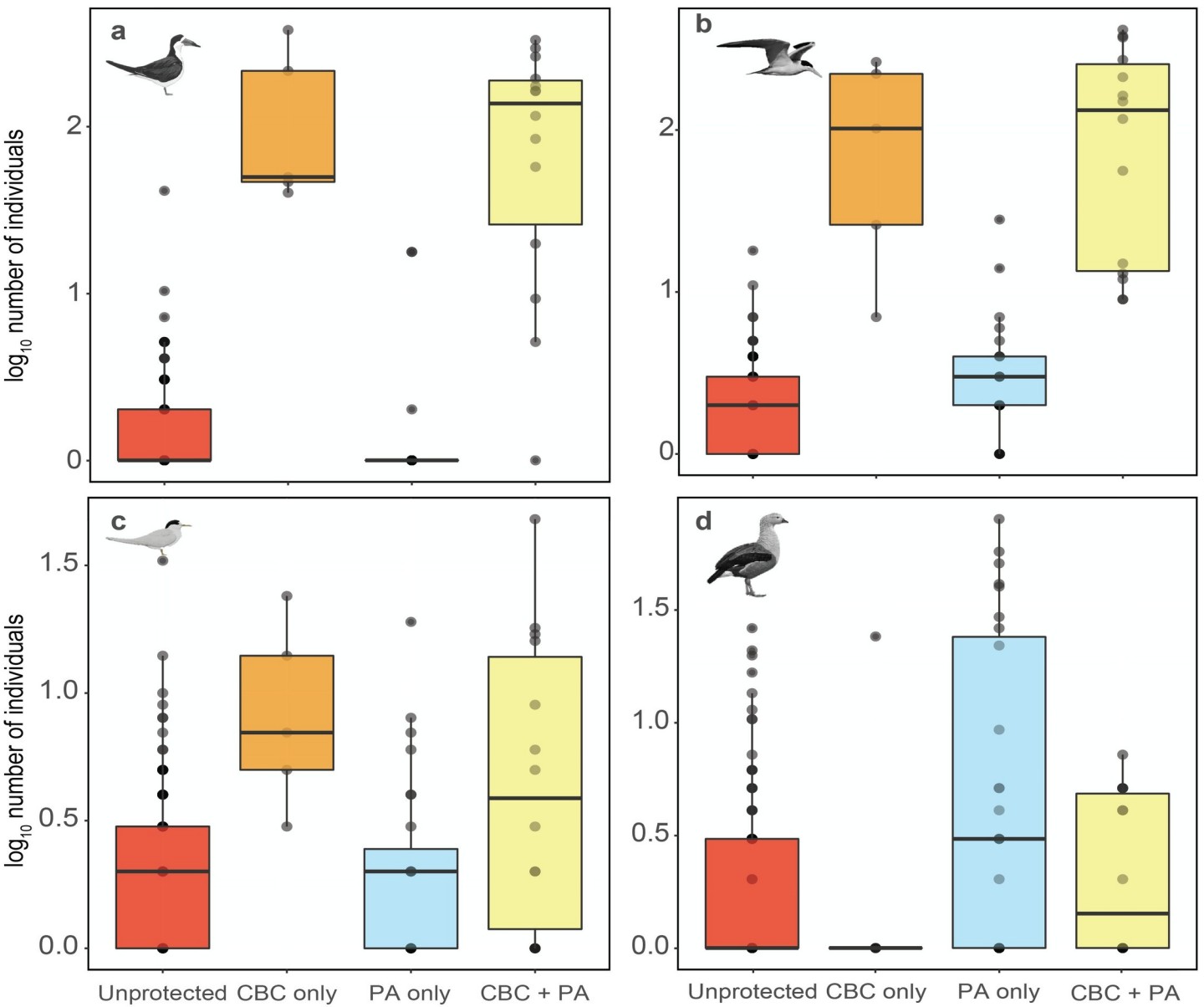

**Fig 3. Box plots showing the number of individuals of four waterbird species surveyed along the Juruá River under different protection regimes.** (A) Black Skimmers; (B) Large-Billed Terns; (C) Yellow-Billed Terns; and, (D) Orinoco Geese. Red, orange blue and yellow boxes represent beaches that are; unprotected; under Community-Based Conservation but outside Protected Areas (CBC-only); inside a PA but without CBC (PA-only); and, under CBC inside a PA (CBC + PA), respectively.

only beaches, compared to unprotected sites. In fact, modelling the Orinoco Goose abundance, PA-only was the strongest predictor, followed by distance to the nearest town (Fig 5D).

## Discussion

### Species responses to site protection

Although Community-Based Conservation programs seemingly align social and conservation outcomes, comprehensive ecological assessments are still lacking [44,45]. In this study, we compared avian populations on fluvial beaches protected by a CBC program that was designed

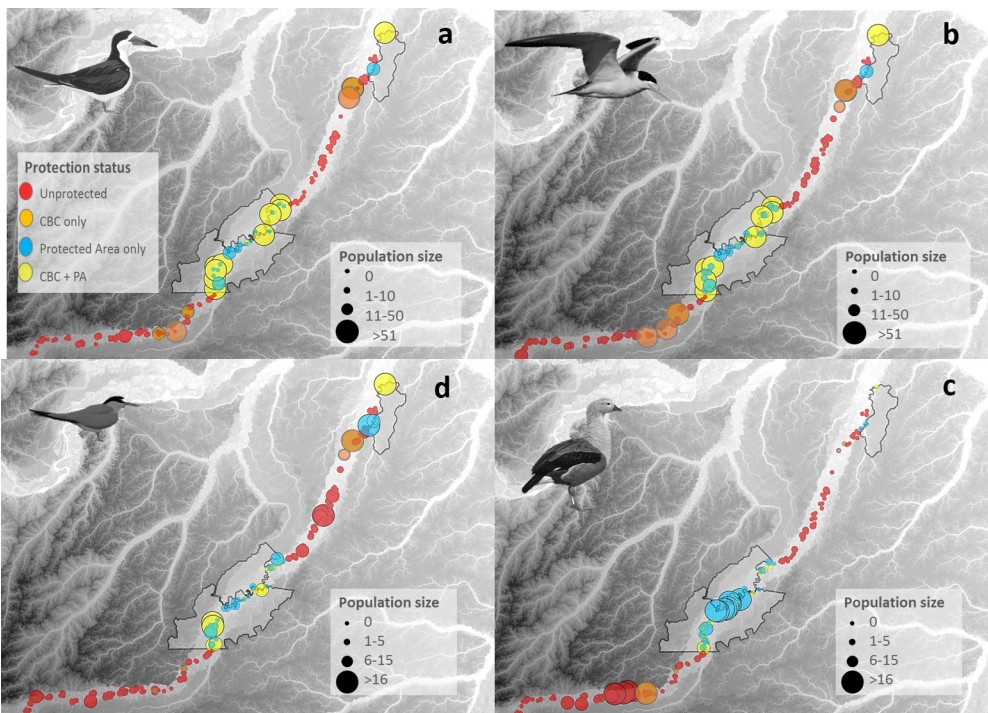

**Fig 4. Distribution map showing the number of individuals of four waterbird species surveyed along ~1,600 km the Juruá River under different protection regimes.** (A) Black Skimmers, (B) Large-Billed Terns, (C) Yellow-Billed Terns, and (D) Orinoco Geese. Symbol sizes are scaled according to census counts; color indicates beach protection status. The background elevation map of the study region illustrates height of local terrain using a color gradient from dark gray (upland) to light gray (lower terrain). Solid black lines represent the boundaries of three sustainable-use PAs. Landsat-7 image courtesy of the U.S. Geological Survey.

to protect turtle-nesting sites inside and outside formal protected areas. Although freshwater turtles are the target of this CBC initiative, the collateral benefits to bird species are impressive. Community-based protection is the most effective strategy to protect migratory waterbirds, independently of whether the arrangement is located inside or outside PAs. These results could have a strong impact on conservation policy in Brazil, considering that CBC arrangements are much easier to establish on the ground, especially under the current political climate where environmental policies and institutions are being dismantled [46].

CBC protection was dramatically important for beach-dependent species. Only 13.5% of our sampled beaches are protected by local communities, but those beaches hosted more than 90% of all Black Skimmer and Large-Billed Tern individuals censused. The effect represents one to two orders of magnitude difference between populations for these species on CBC beaches, compared to beaches lacking CBC enforcement. Although less pronounced, CBC was also important for Yellow-Billed Tern populations. It is likely that this species, which is smaller, less numerous, and more dispersed in the landscape, can succeed nearly equally well on unprotected beaches, as it may not represent a target species for hunting or egg-harvesting.

The Orinoco Goose, an obligate cavity nester, is the only species we assessed that uses fluvial beaches for feeding rather than nesting. We found no effect of CBC protection, although PA status did favor larger populations. The feeding activities of the Orinoco Goose may well be inhibited by the aggressive anti-predator behavior of Black Skimmers and Large-Billed Terns at colony sites [47,48]. Moreover, the activity of nesting birds disturbing and even removing vegetation in the sandy area around nests could represent a direct form of

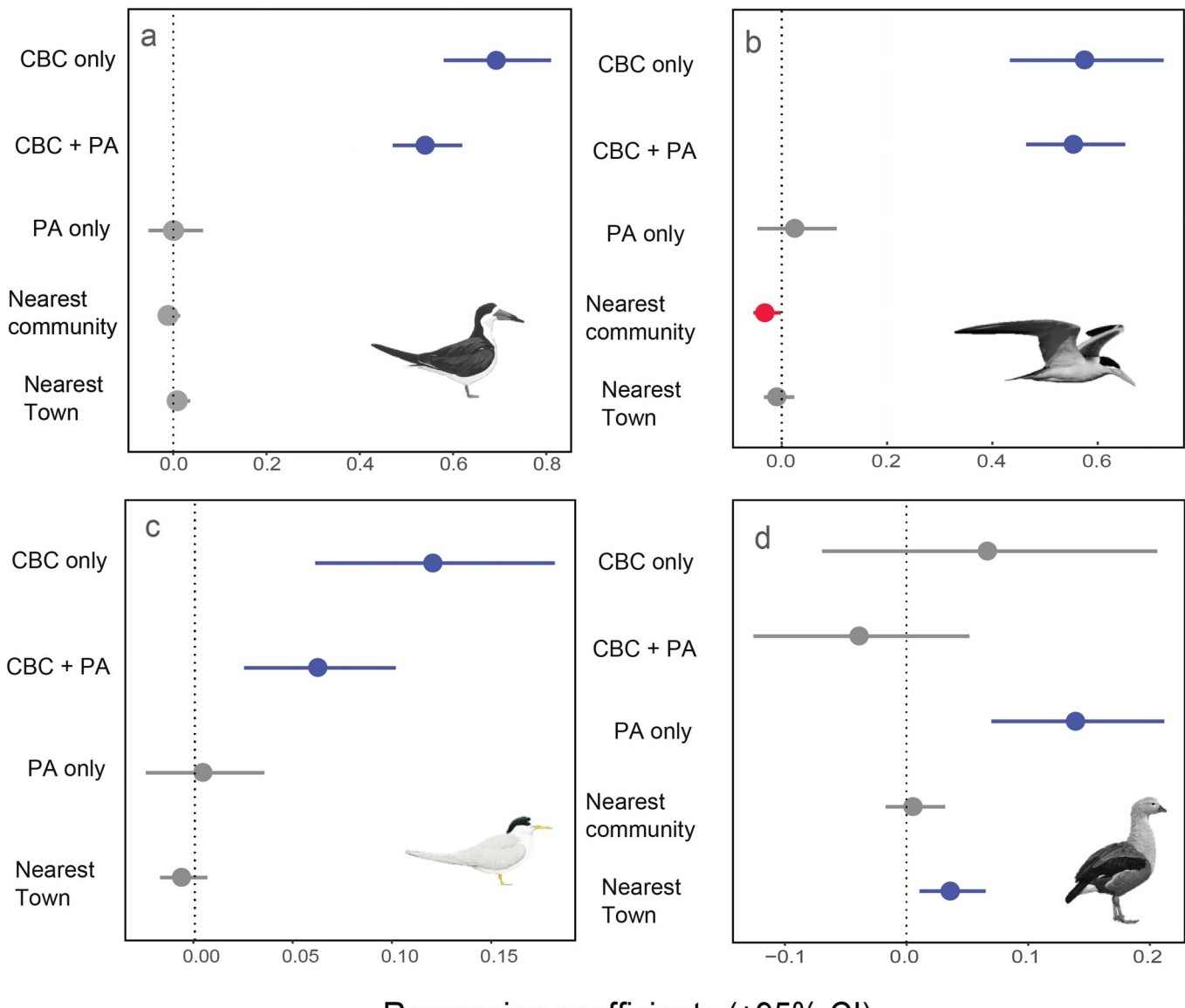

**Fig 5. Coefficient estimates (± 95% confidence intervals) under the linear models, showing the magnitude and direction of effects of different variables on populations of four waterbird species surveyed along the Juruá River.** (A) Black Skimmers, (B) Large-Billed Terns, (C) Yellow-Billed Terns, and (D) Orinoco Geese. For significant variables, the CIs do not cross the vertical dotted line at zero. Blue and red symbols represent positive and negative effects, respectively; gray symbols represent no effect.

competition for food for Orinoco Geese. The Orinoco Goose is also a target species for hunters due to its high food value and the damage it causes to small farmers [30]. In this context, this species may avoid CBC beaches, frequented by local monitors.

Overall, our study reinforces the large collateral benefits afforded by community protection of beaches not only for turtles, but also for multiple non-target species [12]. Colonial species are vulnerable to human disturbance [49], but under CBC protection, these species can breed without the impact of poachers harvesting their eggs. We recognize that we conducted only one survey per beach, and this methodological limitation could lead to a potential error in our estimates through missing or double-counting, considering the potential for individuals to

move between areas over the course of our surveys. However, most of the study species are dependent on beaches and are particularly likely to be site-faithful during the breeding season when we conducted our surveys. Our results are also consistent with previous findings in the same landscape [12] (S2 Fig), increasing confidence in our approach to provide an accurate and realistic estimate for comparative purposes. The orders-of-magnitude differences in nesting activity between CBC and non-CBC beaches suggests that Amazonian beaches suffer from a "shifting baseline" phenomenon, in which our understanding of population ecology for some species is biased by a failure to recognize that historical populations were likely vastly larger than currently observed.

## Patch and landscape effects

Besides CBC protection, we also found that factors such as distance to nearest town and nearest community affected bird populations on our sampled beaches. Distance to local communities plays a context-dependent role in Amazonian waterbird conservation, as the importance of this factor depends on the degree to which a species is a target for subsistence hunters and egg collectors. The effect of distance to the nearest rural community was negative for the abundance of Large-Billed Tern. This pattern has been described in previous studies of other harvested populations, where protected beaches and lakes are more effectively protected if they are in close proximity to local communities, in which more people can enforce community anti-poaching regulations and guard resources [12,50]. Distance to nearest town was an important predictor of Orinoco Goose abundance. This was expected because the town of Carauari hosts a fleet of more than 800 fishing boats, which operate on beaches along the Juruá River and can strongly impact populations of Orinoco Goose, due to the hunting of adults [30].

   We recognize that environmental variables not included in this study could also have an effect on migratory bird abundance [25], and could better explain the existing variance within different protection regimes. However, previous studies in the same landscape have shown that community-protection can overshadow physical environmental variables [12], and any climatic variables are largely consistent within the scale of our study. Therefore, we focused our analysis here on the large-scale effects of different levels of protection status, while recognizing that environmental covariates should continue to be assessed to improve our understanding of the most suitable sites for future expansion of this CBC program.

## Relevance for conservation

There are few examples of community-based conservation accruing benefits to avian species in tropical developing countries. Most examples are focused on participatory monitoring of bird populations, environmental education and capacity building, and the positive outcomes related to increasing data collection through citizen science, public interest and training of local people [51]. On the other hand, collateral benefits of habitat protection of "umbrella species" conservation, while rarely assessed, can generate considerable benefits for a number of species, including non-target bird species that are incidentally protected [12]. The remarkable orders-of-magnitude inadvertent effect of CBC on beach-dependent birds reinforces the highly positive and low-cost role that local communities can have on conservation practices in developing countries. The broad attractiveness of this model is further enhanced by its potential for replication across threatened Amazonian floodplains, ensuring the establishment of protected nesting sites, even if those are located outside formally designated protected areas.

   The most important mechanism behind the effects of CBC arrangement on waterbird populations is the establishment of strict 'no-take' zones maintained over multiple years. In fact, a

well-defined zoning system, including no-take areas, represents a key principle in successful community-based arrangements [52]. Explicit planning of spatio-temporal use zones, including no-take zones between harvesting areas, has the potential to replenish wild populations of harvested resources through source-sink dynamics [31,53,54]. Although community-based protection of beaches is focused on freshwater turtle conservation, which is a natural resource serving a huge socio-cultural value, there are clear collateral benefits at no additional costs for the waterbird populations we censused. Our results emphasize the high potential gains to be made in linking the goals of freshwater turtle and bird conservation, by implementing a low-cost strategy which can be replicated across the Amazon.

## Conclusion

Community-based conservation and management of aquatic resources in Amazonia has been generating clear benefits for both biodiversity conservation and local welfare [55]. These arrangements are much-needed examples of conservation successes which could galvanize stakeholders and policy makers to take bold steps in Amazonian conservation [56]. Ensuring that socioeconomic benefits from a freshwater turtle CBC are successfully delivered to local communities is critical to ensure long-term sustainability [12]. Our findings reinforce the claim that multiple conservation stakeholders should embrace socio-ecological management practices to ensure biodiversity protection. This challenge should be shared more widely with organizations focused on bird conservation to effectively co-create broader multi-taxa conservation programs, since the collateral effects from a freshwater turtle CBC are now shown to play a central role in the conservation of waterbirds at a critical reproductive stage of their life-cycle. This strategy would diversify the range of stakeholders and actors, strengthening the capacity of fundraising and engagement at a local scale.

## Supporting information

**S1 Fig. Testing the effect of time of the day on our surveys of waterbird abundance on fluvial beaches along the Juruá River, western Brazilian Amazonia.** There were no significant ANOVA differences between surveys conducted at 06:30h - 08:00h (Time 1) or 08:30h - 10:00h (Time 2) in any of the surveyed species: (a) *Rynchops niger*, (b) *Phaetusa simplex*, (c) *Sturnella superciliaris*, and (d) *Neochen jubata*. Time of survey was therefore not included in any subsequent models.
(PDF)

**S2 Fig. Testing the effect of year on our surveys of waterbird abundance on fluvial beaches along the Juruá River, western Brazilian Amazonia.** We used a subset of 28 beaches to test (paired t-test) for interannual differences between our surveys conducted in 2016 (this study) and surveys conducted in 2014 (Campos-Silva et al. 2018). Surveys of population size for (a) *Rynchops niger* and (b) *Phaetusa simplex* were consistent at "CBC + PA" and "PA only" beaches in each of these surveys.
(PDF)

**S3 Fig. Testing the effect of protection duration for waterbird abundance on fluvial beaches along the Juruá River, western Brazilian Amazonia.** Population size of target species (a) *Rynchops niger*, $R^2$ = 0.8, p<0.01; (b) *Phaetusa simplex*, $R^2$ = 0.6, p<0.01; (c) *Sturnella superciliaris*, $R^2$ = 0.1, p<0.01; and (d) *Neochen jubata*, $R^2$ = -0.003, p = 0.4) as a function of the number of years that the beach had been under local community protection for unprotected sites (red), PA-only (blue), CBC-only (orange) and CBC + PA (yellow). Unprotected sites and PA-only were left as zero years of protection, considering that in both categories

there is no local site protection by rural communities.
(PDF)

**S1 Table. Abundance of four waterbird species surveyed along the Juruá River, western Brazilian Amazonia.** Full dataset with the number of individuals for each waterbird species per fluvial beach (n = 155), showing protection status, location coordinates, distance to nearest community, distance to nearest town and beach area.
(PDF)

**S2 Table. Model selection explaining the abundance of waterbirds on fluvial beaches along the Juruá River, western Brazilian Amazonia.** Summary of GLM analysis showing the top models (ΔAICc < 2) for each species.
(PDF)

**S1 Methods. Protocol for the community-based protection of freshwater turtles by beach guards.**
(PDF)

## Acknowledgments

We thank the Secretaria de Estado do Meio Ambiente Amazonas (SEMA-DEMUC) and Instituto Chico Mendes de Conservação da Biodiversidade (ICMBio) for authorizing the research, and two anonymous reviewers for comments on an earlier version of the manuscript. We are grateful to Projeto Pé-de-Pincha at Universidade Federal do Amazonas and all local communities along the Juruá River for their support. We thank José Alves (Silas) for their assistance in the field. We also thank Hugo C.M. Costa, Renato Cintra and Whaldener Endo for photos c, e and g within Fig 1. This publication is part of the Instituto Juruá series (www.institutojurua.org.br).

## Author Contributions

**Conceptualization:** João Vitor Campos-Silva, Carlos A. Peres, Joseph E. Hawes, Paulo C. M. Andrade, Lisa Davenport.

**Data curation:** João Vitor Campos-Silva, Carlos A. Peres.

**Formal analysis:** João Vitor Campos-Silva, Carlos A. Peres, Joseph E. Hawes, Mark I. Abrahams, Lisa Davenport.

**Funding acquisition:** João Vitor Campos-Silva, Carlos A. Peres, Joseph E. Hawes.

**Investigation:** João Vitor Campos-Silva, Lisa Davenport.

**Methodology:** João Vitor Campos-Silva, Joseph E. Hawes, Mark I. Abrahams, Lisa Davenport.

**Project administration:** João Vitor Campos-Silva, Carlos A. Peres.

**Resources:** João Vitor Campos-Silva, Carlos A. Peres, Joseph E. Hawes.

**Software:** João Vitor Campos-Silva.

**Supervision:** Carlos A. Peres.

**Validation:** João Vitor Campos-Silva, Carlos A. Peres, Joseph E. Hawes, Lisa Davenport.

**Visualization:** João Vitor Campos-Silva, Carlos A. Peres, Joseph E. Hawes, Lisa Davenport.

**Writing – original draft:** João Vitor Campos-Silva.

**Writing – review & editing:** João Vitor Campos-Silva, Carlos A. Peres, Joseph E. Hawes, Mark I. Abrahams, Paulo C. M. Andrade, Lisa Davenport.

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
