## [Decision Letter · Decision Letter 0]

2 Aug 2020

PONE-D-20-19231

Community-based conservation with formal protection provides large collateral benefits to migratory waterbirds of Amazonia

PLOS ONE

Dear Dr. Campos-Silva,

Thank you for submitting your manuscript to PLOS ONE. After careful consideration, we feel that it has merit but does not fully meet PLOS ONE’s publication criteria as it currently stands. Therefore, we invite you to submit a revised version of the manuscript that addresses the points raised during the review process.

Please submit your revised manuscript within 60 days. If you will need more time than this to complete your revisions, please reply to this message or contact the journal office at plosone@plos.org. Please include the following items when submitting your revised manuscript:

We look forward to receiving your revised manuscript.

Kind regards,

Yong Zhang

Academic Editor

PLOS ONE

Journal Requirements:

2. In your Methods section, please provide additional location information of the study sites, including geographic coordinates for the data set if available.

3.Thank you for stating the following in the Acknowledgments Section of your manuscript:

[This study was funded by a Darwin Initiative for the Survival of Species grant (DEFRA, UK; Ref. 20-001) awarded to C.A.P., a CAPES PhD scholarship (Ref. 1144985) and CAPES postdoctoral grant (Ref. 1666302) to J.V.C.S., a CAPES postdoctoral grant (Ref. 1530532) and internal funding from Anglia Ruskin University to J.E.H., and National Geographic Society, Grant # WW1-277R-18 to L.D.t]

 [The funders had no role in study design, data collection and analysis, decision to publish, or preparation of the manuscript.]

4.We note that [Figure(s) 1 and 4] in your submission contain [map/satellite] images which may be copyrighted. All PLOS content is published under the Creative Commons Attribution License (CC BY 4.0), which means that the manuscript, images, and Supporting Information files will be freely available online, and any third party is permitted to access, download, copy, distribute, and use these materials in any way, even commercially, with proper attribution. For these reasons, we cannot publish previously copyrighted maps or satellite images created using proprietary data, such as Google software (Google Maps, Street View, and Earth). For more information, see our copyright guidelines: http://journals.plos.org/plosone/s/licenses-and-copyright.

1.    You may seek permission from the original copyright holder of Figure(s) [1 and 4] to publish the content specifically under the CC BY 4.0 license. 

Additional Editor Comments (if provided):

Although both reviewers think that the topic is interesting, they also offered some main comments which i think are reasonably. The main concerns are about survey methods and data analyses, as pointed by the two reviewers, see comments below.

Reviewers' comments:

Reviewer's Responses to Questions

**Comments to the Author**

1. Is the manuscript technically sound, and do the data support the conclusions?

Reviewer #1: No

Reviewer #2: Partly

2. Has the statistical analysis been performed appropriately and rigorously? 

Reviewer #1: No

Reviewer #2: Yes

3. Have the authors made all data underlying the findings in their manuscript fully available?

Reviewer #1: Yes

Reviewer #2: Yes

4. Is the manuscript presented in an intelligible fashion and written in standard English?

Reviewer #1: Yes

Reviewer #2: Yes

5. Review Comments to the Author

Reviewer #1: I read this manuscript with great interest. The topic is very worthy of study given that the Amazon is a major biodiversity hotspot yet the effectiveness of different types of protected areas has received limited attention. At face value the conclusions offer valuable new information on how community-led conservation efforts can have major benefits for wildlife, not only for the target species (in the case of CBCs, freshwater turtles), but also waterbird species which were the focus of your study. That said, the value of conservation research is dependent on appropriate study design and analyses. Having read your study, I had a number of concerns about the data collection and analyses:

(i) I was not clear on why the statistical analyses are done in two steps, the first which looks just at the protected area designation, and the second which attempts to account for covariate effects. I do not see why these could not have been combined into a single analysis that tested for differences in abundance between different protected area designations whilst accounting for key covariates such as beach area. A single analysis would have been optimal. The first analysis in particular, did not appear to test the assumptions correctly (normality is assumed in the residuals, not the raw data, and the other assumptions also need to be assessed), whilst not accounting for factors such as beach area which could have affected bird counts. In my view the first analysis could be dropped and the analyses refocused on the second analysis.

(ii) There is likely to be an issue with spatial autocorrelation in your data set, that needs to be accounted for within your analysis. Figure 1 shows clearly that many of your sites of each designation type (e.g. CBC) are clustered together. This is often inevitable in real-world conservation situations, but it presents a problem when analyzing such data because sites that are closer together are more likely to have similar waterbird abundances than sites which are further apart. They are also more likely to be more similar in terms of their covariate values. As a first step, you should use a metric such as Moran's I to quantify the strength of spatial autocorrealtion in your data set. There are a number of R packages that allow you to estimate Moran's I and test its significance, such as the spdep package. If spatial autocorrelation is very weak (i.e. non-significant) you may be ok with your original analysis, but if it is significant you will need to account for the autocorrelation in order to achieve accurate and robust conclusions. The following two papers provide examples of how this could be achieved: Beale CM, Lennon JJ, Yearsley JM, Brewer MJ, Elston DA. 2010. Regression analysis of spatial data. Ecology Letters 13:246-264. Dormann CF, McPherson JM, Araújo MB, Bivand R, Bolliger J, Carl G, Davies RG, Hirzel A, Jetz W, Kissling WD. 2007. Methods to account for spatial autocorrelation in the analysis of species distributional data: a review. Ecography 30:609-628.

(iii) There needs to be more explicit consideration of the process of designation of the different types of protected area status on the beaches. Currently, the manuscript attempts to link the higher abundances in certain types of site (e.g. CBC) with the protection status of the site. But could it be the case that these areas already had higher abundances of waterbirds when they were designated, rather than having achieved higher abundances because of the status? Especially if the sites were chosen because they were high quality sites for turtles, could it be that they were also inherently high quality sites for waterbirds? Globally, it is frequently the case that sites of high abundance are designated as protected areas for that species, which complicates assessments of the effects of protected areas. At the very least, your manuscript ought to discuss this issue and clarify what you think the mechanisms are through which CBC sites come to have higher waterbird abundances, and what the evidence is for these mechanisms (keeping in mind that in this paper you report only waterbird abundance estimates taken some time after CBC designation).

(iv) I also had some concerns about the waterbird survey methodologies. I appreciate that surveying large areas in difficult terrain is logistically challenging, but some consideration needs to be given to the issue of birds that could have been missed or double-counted. This is because the surveys of the 155 beaches were not synchronous; given the length of the study areas and the movement speed that you report, the surveys were undertaken over the course of many days (it would be useful for you to clarify the exact time period in August 2016, in terms of numbers of days). Given what is known about the 4 species, it is possible that individuals moved between sites over the course of your surveys (especially if the surveys disturbed birds from the beaches), resulting in birds that were either missed or double-counted? As your study was carried out in the breeding season, the birds may have been largely site-faithful, but given the close proximity of many of your sites, I think that at the least you should mention these issues in either the methods or discussion, along with your thoughts on whether these issues could have affected your count data.

(v) My other concern about the waterbird survey methodology was the lack of standardization of the timing of the surveys. For diurnally-active bird species, activity patterns can show consistent patterns across daylight hours (for example, see: Robbins, C.S. 1981. Effect of time of day on bird activity. Studies in avian biology, 6, 275-286). The fluctuations in activity patterns can result in detection probabilities that vary with time of day (Palmeirim, J.M. & Rabaça, J.E. 1994. A Method to Analyze and Compensate for Time-of-Day Effects on Bird Counts. Journal of Field Ornithology, 65, 17-26). Therefore, in your study the abundance of birds on a beach could have been affected by the time during the day that the count was conducted. As your counts started early in the day, could this have included some individuals that would have subsequently left their nest area to forage elsewhere, i.e. would these early counts have recorded lower abundance estimates if they'd been made in the middle of the day? If you have recorded the time of each survey, you could account for time-of-day (perhaps in blocks of time, e.g. 06:30-09:30, 09:30-12:30, and so on) within your models.

I also have a number of additional, more specific comments, which I hope will help you to improve your study:

Page 3, Lines 44-46: Here, I recommend also adding the following classic paper on tropical freshwater ecosystems, which highlights many of the issues that these ecosystems face: Dudgeon, D. (1992). Endangered ecosystems: a review of the conservation status of tropical Asian rivers. Hydrobiologia, 248, 167-191.

Page 3, Lines 56-62: Do the CBCs allow some limited exploitation of turtles by local communities, or is no exploitation allowed? The information in the next paragraph suggests that the latter is true, but it would be good to clarify this point to help the reader understand how the system works.

Page 5, Line 105: The "-" is not needed before "km" and so can be deleted.

Page 8, Line 168: Do you mean 20 birds in total (of all species) or 20 individuals of a single species?

Page 8, Lines 179-182: Was Kruskal-Wallis used here as a non-parametric alternative to ANOVA? Crucially, ANOVA assumes normality of the residuals, not the raw data, so what you've done here is not the correct procedure. You also have not mentioned any other test assumptions beyond normality. ANOVA is a more powerful test than Kruskal-Wallis and ought to be used if the asssumptions are met.

Page 8, Line 180: Typo -> "Kruskal" not "Kruskall".

Page 9, Line 203: More detail is needed here. Specifically, which assumptions did you check and how did you do this? Did you use visual inspections of residuals or formal statistical tests? Also, how did you test for collinearity? Did you use Variance Inflation Factors or correlations? These results should be reported in the manuscript.

Page 12, Line 245: Do the P values in the table (and elsewhere in the manuscript) take account of the fact that multiple p values have been calculated? Using Holm-Bonferroni corrections (or similar) would resolve this issue.

Page 12, Lines 254-256: The results reported here do not appear to match the values reported in Table 1. For example, the text states that "mean population size of Black Skimmer was 135-fold and 108-fold larger on CBC only and CBC + PA beaches than at unprotected sites, and only 1.3 higher in PA only beaches compared with unprotected sites", but the data in Table 1 show that population size on CBC only beaches is 134 vs 1 at unprotected sites, 107.2 at CBC + PA sites, and 1.2 at PA only versus 1.0 at unprotected sites. I recommend checking and revising the entire results section so that the values reported in the text agree with the values reported in the tables and figures.

Page 15, Line 305: As per major comment iii above, I think that there needs to be a more nuanced interpretation of your results. You do not have time series data to show how waterbird abundances have changes pre- and post-designation as a CBC site (or as any other type of protected area) so you cannot conclude that species have "responded" to site protection. As mentioned earlier, it may be the case that these sites already had high waterbird abundances when they were designated as CBC sites, or it could be that the protection afforded to CBC sites has resulted in the higher abundances. Unless you have the data to distinguish between these two hypotheses, you need to be more cautious in how your frame your findings.

Page 16, Lines 346-347: Can you cite any evidence to support the idea that smaller beaches have lower fishing activity?

Page 25, Lines 532-534: Which version number of R did you use in your analyses? Please add that information here.

Reviewer #2: In this paper, Campos-Silva et al monitor the population of four waterbirds on 155 beaches along the Juruá River and find that the waterbirds populations are significantly larger on the CBC beaches than that on the other beaches. Therefore, Campos-Silva et al declare that the collateral benefits from local community’s conservation activities on turtles explain why the four waterbirds aggregate on the CBC beaches using regression analysis. This is an important area of biodiversity conservation research, particularly in a biodiversity hotspot region with high level of human disturbance.

I have two comments that I believe need addressing before publication.

My first concern is the authors monitor the waterbirds population only one time. Due to the strong ability and large range of birds flying, it is doubtful whether only one survey can truly reflect the actual population numbers of the four birds.

My second concern is that the collateral benefits from community conservation is not fully proven in the manuscript. Bird population is affected by a variety of environmental factors, including temperature, precipitation, food resources, etc., which have been confirmed in a series of literatures. In lines 354-355, the authors quoted their previous research to show that community conservation overshadow other environmental factors, which is difficult to convince readers. Figure 4 shows that most of the beaches where waterbirds congregate are distributed in the middle of the Juruá River. Therefore, it is also possible that the longitude, latitude, temperature or food resources determine the bird population, while CBC happens to be run in the suitable beaches for waterbirds. Furthermore, the high SD values of Black Skimmer and Large-Billed Tern on the PA only beaches somewhat demonstrate that some environmental factors other than CBC mainly influence waterbirds populations. Therefore, it is necessary for the author to take more environmental parameters into account when construct the regression models to verify the conclusion. If the strengthened model still shows that CBC is the dominant factor, the paper will be more convincing. If the conclusion of the strengthened model is not consistent with the current one, the manuscript needs to make corresponding modifications before publication.

6. PLOS authors have the option to publish the peer review history of their article (what does this mean?). If published, this will include your full peer review and any attached files.

Reviewer #1: No

Reviewer #2: No

---

## [Author Response · Author response to Decision Letter 0]

5 Dec 2020

We thank the Associate Editor and reviewers for their constructive comments. We were happy to accept every comment, and are confident that the manuscript has become stronger and clearer as a result. Please see all details of our responses in our point-by-point responses in attach. We did not pasted all responses here, because there are some figures to illustrate oure responses that are not allowed in this box.

Best wishes

Joao Campos-Silva

---

## [Decision Letter · Decision Letter 1]

22 Mar 2021

PONE-D-20-19231R1

Community-based conservation with formal protection provides large collateral benefits to Amazonian migratory waterbirds

PLOS ONE

Dear Dr. Campos-Silva,

Thank you for submitting your manuscript to PLOS ONE. After careful consideration, we feel that it has merit but does not fully meet PLOS ONE’s publication criteria as it currently stands. Therefore, we invite you to submit a revised version of the manuscript that addresses the points raised during the review process.

Both reviewers feel your manuscript will be a valuable contribution to the field.  That said, Reviewer 1 has suggested a few additional, but minor revisions to clarify or expand upon certain points.  None of these should require much effort on your part and will likely broaden the relevance and readership of your publication.

We look forward to receiving your revised manuscript.

Kind regards,

Janice L. Bossart

Academic Editor

PLOS ONE

Journal Requirements:

Reviewers' comments:

Reviewer's Responses to Questions

**Comments to the Author**

1. If the authors have adequately addressed your comments raised in a previous round of review and you feel that this manuscript is now acceptable for publication, you may indicate that here to bypass the “Comments to the Author” section, enter your conflict of interest statement in the “Confidential to Editor” section, and submit your "Accept" recommendation.

Reviewer #1: (No Response)

Reviewer #3: All comments have been addressed

2. Is the manuscript technically sound, and do the data support the conclusions?

Reviewer #1: Yes

Reviewer #3: Yes

3. Has the statistical analysis been performed appropriately and rigorously? 

Reviewer #1: Yes

Reviewer #3: Yes

4. Have the authors made all data underlying the findings in their manuscript fully available?

Reviewer #1: Yes

Reviewer #3: Yes

5. Is the manuscript presented in an intelligible fashion and written in standard English?

Reviewer #1: Yes

Reviewer #3: Yes

6. Review Comments to the Author

Reviewer #1: Thank you for the thorough and well-documented revisions that you have carried out. I think that you have addressed all of our points and the manuscript has been greatly improved. Reading through your revised text, I had a few final comments and suggestions, which I hope will prove useful:

Lines 48-50. I agree, the region has such high biodiversity value but faces a range of anthropogenic threats. In addition to the reference you already give, I would recommend also citing this subsequent paper which provides the latest perspective on this topic: Castello, L. & Macedo, M.N. (2016). Large‐scale degradation of Amazonian freshwater ecosystems. Global Change Biology, 22, 990-1007.

Line 51. Can you clarify the phrase “most comprehensive protected area systems” to explain in what way(s) the protected area system is comprehensive? In terms of area? Number of sites? Number of restrictions on anthropogenic activities? Or something else?

Lines 87-88. “Increasing and strengthening conservation strategies both inside and outside PA boundaries is therefore imperative to maintaining migratory bird populations.” I think that this idea, which is fundamental to the rationale of your study, should be developed further before the readers moves to the next paragraph. Specifically, I think that you should highlight the need for waterbird monitoring, as that it critical to efforts to diagnose and alleviate anthropogenic threats to waterbird populations. Without monitoring of waterbird populations, we would lack the information necessary to undertake protective measures, such as designating protected areas or regulating anthropogenic activities that impact populations. here are two recent examples that illustrate the point: Wood, K.A., Brown, M.J., Cromie, R.L., Hilton, G.M., Mackenzie, C., Newth, J.L., Pain, D.J., Perrins, C.M. & Rees, E.C. (2019). Regulation of lead fishing weights results in mute swan population recovery. Biological Conservation, 230, 67-74. Martínez-Abraín, A., Jiménez, J., Gómez, J.A. & Oro, D. (2016). Differential waterbird population dynamics after long-term protection: The influence of diet and habitat type. Ardeola, 63, 79-101. I think that these few extra lines and references would help to steer the reader towards appreciating why waterbird monitoring is needed and what could be gained from this information, which would give the reader a greater appreciation for your study.

Lines 394-397. “This challenge should be shared more widely with organizations focused on bird conservation, since the collateral effects from a freshwater turtle CBC are now shown to play a central role in the conservation of waterbirds at a critical reproductive stage of their lifecycle.” Can you develop your recommendation further here, for example by giving recommendations for how the challenge could be shared effectively? Conservation projects with broader suits of biodiversity and social development targets, with stakeholders from across a more diverse range of stakeholders? Monitoring programmes undertaken by multiple stakeholder groups?

Reviewer #3: Campos-Silva et al. provide compelling evidence that river areas protected local communities for turtle nesting provide additional benefits for colonial nesting waterbirds, but not for a solitary foraging goose species. The authors compared and surveyed three areas: unprotected, government designated protected areas (PAs) and community-based conservation (CBC) areas in the Amazon. The spatial coverage was significant: 1600 km and 155 sites. The analyses appeared to address the issue of many sites that had no nesting waterbirds (i.e., zeros). What is interesting about this study is the findings that CBC programs, implemented for a different taxon (turtles), had an umbrella positive benefit for all three species of colonial nesting waterbirds, but not for the Orinoco goose. The analyses made sense and their conclusions, that CBC programs can have added benefits to other species, were based on a strong study design. While there is recent discussion and push to involve local communities in conservation initiatives, there are few data that evaluate that management option, thus this study is a valuable contribution.

Specific comments

Line 69 beach guards (CBC) – Is there a formal protocol that the beach guards follow beyond what was described in the Introduction? If yes, could it be shared under Supplemental section? I wonder if for example, they are permitted to have dogs, or if dogs are prohibited? The presence of dogs would presumably negatively impact ground nesting waterbirds.

Line 161 waterbird surveys - Were birds flushed during the survey? If yes, is it possible they relocated to another area being surveyed? (i.e., confirm no double counting within surveys happening simultaneously).

Line 209 Results - While the beach area for each survey site is presented in the supplemental data, it would be good to provide some descriptive stats (area per category) to get a sense of the distribution of beach area for Unprotected, PA and CBC.

7. PLOS authors have the option to publish the peer review history of their article (what does this mean?). If published, this will include your full peer review and any attached files.

Reviewer #1: No

Reviewer #3: No

---

## [Author Response · Author response to Decision Letter 1]

24 Mar 2021

Reviewer #1: Thank you for the thorough and well-documented revisions that you have carried out. I think that you have addressed all of our points and the manuscript has been greatly improved. Reading through your revised text, I had a few final comments and suggestions, which I hope will prove useful.

Authors: We appreciate the reviewer’s effort in improving our manuscript, and are grateful for all their comments and suggestions.

Lines 48-50. I agree, the region has such high biodiversity value but faces a range of anthropogenic threats. In addition to the reference you already give, I would recommend also citing this subsequent paper which provides the latest perspective on this topic: Castello, L. & Macedo, M.N. (2016). Large‐scale degradation of Amazonian freshwater ecosystems. Global Change Biology, 22, 990-1007.

Authors: Done. Citation added to line 48.

Line 51. Can you clarify the phrase “most comprehensive protected area systems” to explain in what way(s) the protected area system is comprehensive? In terms of area? Number of sites? Number of restrictions on anthropogenic activities? Or something else?

Authors: The Brazilian protected area (PA) system is composed of 2201 PAs covering more than 250 million hectares. We clarified this point in lines 51-52.

Lines 87-88. “Increasing and strengthening conservation strategies both inside and outside PA boundaries is therefore imperative to maintaining migratory bird populations.” I think that this idea, which is fundamental to the rationale of your study, should be developed further before the readers moves to the next paragraph. Specifically, I think that you should highlight the need for waterbird monitoring, as that it critical to efforts to diagnose and alleviate anthropogenic threats to waterbird populations. Without monitoring of waterbird populations, we would lack the information necessary to undertake protective measures, such as designating protected areas or regulating anthropogenic activities that impact populations. here are two recent examples that illustrate the point: Wood, K.A., Brown, M.J., Cromie, R.L., Hilton, G.M., Mackenzie, C., Newth, J.L., Pain, D.J., Perrins, C.M. & Rees, E.C. (2019). Regulation of lead fishing weights results in mute swan population recovery. Biological Conservation, 230, 67-74. Martínez-Abraín, A., Jiménez, J., Gómez, J.A. & Oro, D. (2016). Differential waterbird population dynamics after long-term protection: The influence of diet and habitat type. Ardeola, 63, 79-101. I think that these few extra lines and references would help to steer the reader towards appreciating why waterbird monitoring is needed and what could be gained from this information, which would give the reader a greater appreciation for your study.

Authors: Thank you very much for your suggestion. We have now highlighted in the text the need for strengthened monitoring of waterbird populations (lines 88-93), and included the suggested references. 

Lines 394-397. “This challenge should be shared more widely with organizations focused on bird conservation, since the collateral effects from a freshwater turtle CBC are now shown to play a central role in the conservation of waterbirds at a critical reproductive stage of their lifecycle.” Can you develop your recommendation further here, for example by giving recommendations for how the challenge could be shared effectively? Conservation projects with broader suits of biodiversity and social development targets, with stakeholders from across a more diverse range of stakeholders? Monitoring programmes undertaken by multiple stakeholder groups?

Authors: Authors: We have developed our recommendation section within the Conclusion, as suggested (lines 402 to 409).

Reviewer #3: Campos-Silva et al. provide compelling evidence that river areas protected local communities for turtle nesting provide additional benefits for colonial nesting waterbirds, but not for a solitary foraging goose species. The authors compared and surveyed three areas: unprotected, government designated protected areas (PAs) and community-based conservation (CBC) areas in the Amazon. The spatial coverage was significant: 1600 km and 155 sites. The analyses appeared to address the issue of many sites that had no nesting waterbirds (i.e., zeros). What is interesting about this study is the findings that CBC programs, implemented for a different taxon (turtles), had an umbrella positive benefit for all three species of colonial nesting waterbirds, but not for the Orinoco goose. The analyses made sense and their conclusions, that CBC programs can have added benefits to other species, were based on a strong study design. While there is recent discussion and push to involve local communities in conservation initiatives, there are few data that evaluate that management option, thus this study is a valuable contribution.

Authors: We are grateful for Rev#3’s very positive comments on our manuscript, including its novelty value in highlighting rare evidence of the conservation co-benefits accrued to non-target species.

Specific comments

Line 69 beach guards (CBC) – Is there a formal protocol that the beach guards follow beyond what was described in the Introduction? If yes, could it be shared under Supplemental section? I wonder if for example, they are permitted to have dogs, or if dogs are prohibited? The presence of dogs would presumably negatively impact ground nesting waterbirds.

Authors: We have included the main obligations of beach guards in the Supporting Information (S1 Methods). Although there is no direct mention regarding the presence of domestic and feral dogs in protected beaches, their access to beaches is informally banned because they also represent a clear depredation threat for nesting turtles, as well as waterbirds.

Line 161 waterbird surveys - Were birds flushed during the survey? If yes, is it possible they relocated to another area being surveyed? (i.e., confirm no double counting within surveys happening simultaneously).

Authors: Waterbirds are habituated to the presence of beach guards who conducted surveillance of the beaches on foot on a daily basis. Our sampling was conducted at a distance, so our presence did not scare the birds away. In addition, since most of our target waterbird species use the beaches to breed and exhibit a great deal of nest-site fidelity, rather than for other activities (e.g. foraging), we are confident that our surveys provide a robust estimate of breeding bird population sizes on each beach. We clarified this point in the Methods (lines 183-185).

Line 209 Results - While the beach area for each survey site is presented in the supplemental data, it would be good to provide some descriptive stats (area per category) to get a sense of the distribution of beach area for Unprotected, PA and CBC.

Authors: Thank-you. We agree with this suggestion and have inserted the mean area and standard deviation for each category of beach protection in the first paragraph of our Results (lines 228-231).

---

## [Editor Report · Decision Letter 2]

30 Mar 2021

Community-based conservation with formal protection provides large collateral benefits to Amazonian migratory waterbirds

PONE-D-20-19231R2

Dear Dr. Campos-Silva,

We’re pleased to inform you that your manuscript has been judged scientifically suitable for publication and will be formally accepted for publication once it meets all outstanding technical requirements.

Kind regards,

Janice L. Bossart

Academic Editor

PLOS ONE
---

## [Editor Report · Acceptance letter]

1 Apr 2021

PONE-D-20-19231R2 

Community-based conservation with formal protection provides large collateral benefits to Amazonian migratory waterbirds 

Dear Dr. Campos-Silva:

I'm pleased to inform you that your manuscript has been deemed suitable for publication in PLOS ONE. Congratulations! Your manuscript is now with our production department. 

Kind regards, 

on behalf of

Dr. Janice L. Bossart 

Academic Editor

PLOS ONE